# A Novel Method for the Preparation of Casein–Fucoidan Composite Nanostructures

**DOI:** 10.3390/polym16131818

**Published:** 2024-06-27

**Authors:** Nikolay Zahariev, Bissera Pilicheva

**Affiliations:** 1Department of Pharmaceutical Sciences, Faculty of Pharmacy, Medical University of Plovdiv, 15A Vassil Aprilov Blvd, 4002 Plovdiv, Bulgaria; nikolay.zahariev@mu-plovdiv.bg; 2Research Institute at Medical University of Plovdiv, 15A Vassil Aprilov Blvd, 4002 Plovdiv, Bulgaria

**Keywords:** casein, fucoidan, polyelectrolyte complexation, spray drying, experimental design, nanocomposites

## Abstract

The aim of the study was to develop casein–fucoidan composite nanostructures through the method of polyelectrolyte complexation and subsequent spray drying. To determine the optimal parameters for the preparation of the composite structures and to investigate the influence of the production and technological parameters on the main structural and morphological characteristics of the obtained structures, 3(k-p) fractional factorial design was applied. The independent variables (casein to fucoidan ratio, glutaraldehyde concentration, and spray intensity) were varied at three levels (low, medium, and high) and their effect on the yield, the average particle size, and the zeta potential were evaluated statistically. Based on the obtained results, models C1F1G1Sp.30, C1F1G2Sp.40, and C1F1G3Sp.50, which have an average particle size ranging from (0.265 ± 0.03) µm to (0.357 ± 0.02) µm, a production yield in the range (48.9 ± 2.9) % to (66.4 ± 2.2) %, and a zeta potential varying from (−20.12 ± 0.9) mV to (−25.71 ± 1.0) mV, were selected as optimal for further use as drug delivery systems.

## 1. Introduction

In recent decades, biopolymers have been the subject of a serious scientific interest for the development of nanosized drug delivery systems due to their specific characteristics and numerous advantages over synthetic polymers, such as low immunogenicity, high biocompatibility, biodegradability, and low toxicity [1]. Biopolymers such as pep-tides and polysaccharides have been widely used for the preparation of nanoparticulate drug delivery systems that can provide controlled and targeted release, improving biodistribution of the encapsulated drug and minimizing its systemic toxicity [2]. Compared to polysaccharides, proteins are preferred as structure-forming biopolymers for the production of drug delivery systems due to their ability to form smaller-in-size particles [3]. Among the proteins used for the formulation of nanosized carriers, casein has shown optimal structural and physicochemical properties [4]. Casein is a collective term for a family of calcium (phosphate)-binding phosphoproteins commonly found in mammalian milk. Bovine milk casein consists of four peptides, namely αs1, αs2, β, and k, which differ in amino acid, phosphorus, and carbohydrate content, but all of them possess amphiphilic properties [5]. Cysteine amino acid residues, which are involved in the formation of disulfide bonds, are found only in the polypeptide chains of k-casein. The lack of secondary structures of the casein molecules, due to the proline-rich amino acid residues and the ability to bind to calcium phosphate, leads to the formation of electrostatic, hydrogen, and hydrophobic interactions [6]. As a result, self-association of the casein peptides occurs, resulting in the formation of stable agglomerates known as casein micelles [6]. The inner part of the micelle is composed of αs1, αs2, and β caseins, while the outer layer, which stabilizes the micelle, contains glycosylated k-casein [7]. Casein micelles exhibit pH-dependent behavior. A decrease in the surface negative charge of the casein micelles leads to their contraction, while an increase in the charge leads to an electrostatic repulsion of the casein molecules [8].

A major problem in the development of casein-based nanostructures is the low drug encapsulation efficiency and drug loading index [9]. This is related to the use of large amounts of drug delivery systems to ensure the desired dose of the incorporated therapeutic agent [9]. Thus, the development of suitable technological approaches to overcome these drawbacks is necessary for the preparation of stable and effective drug delivery systems. One of the main approaches is the formation of casein–polysaccharide complexes.

Due to its specific physicochemical properties, casein can participate in the formation of composite nanostructures because of polyelectrolyte or chemical interactions with one or more polymers. Compared to chemically and physically cross-linked casein nanoparticles, composite nanoparticles are characterized by higher stability and the possibility of modified drug release [10].

Protein-based carriers for drug delivery, including casein, are characterized by physical instability, resulting in protein aggregation and precipitation at a pH close to the isoelectric point of the protein. The incorporation of polysaccharides into the carrier structure induces steric hindrance, which limits protein aggregation [11]. As a result, protein–polysaccharide composite nanoparticles have increased stability and more efficiently encapsulate hydrophilic and hydrophobic therapeutic agents [12]. These complexes also provide protection to the included substance from oxidation and hydrolysis under the influence of environmental factors such as pH, light, and temperature [13]. In addition, the colloidal network composing the nanoparticles, built of protein–polysaccharide complexes, provides a delayed release of the incorporated therapeutic agent [14]. Polysaccharide complexation of proteins can be due to both covalent and non-covalent binding of biopolymers. Covalent protein–polysaccharide conjugation is due to enzymatic cross-linking (oxidases and transglutaminases), chemical cross-linking (genipin, glutaraldehyde, and polyethylene glycol), or through the Maillard reaction [15].

In a study conducted by Pan et al., the Maillard reaction was used to prepare casein–dextran copolymer [16]. The obtained structures showed pH-dependent behavior, with micelle formation occurring at a pH equal to the isoelectric point of casein, and demonstrated successful incorporation of the fluorescent agent pyrene [16]. 

Covalent interaction between proteins and polysaccharides also occurs because of cross-linking with transglutaminase, which catalyzes acyl-transfer reactions between the γ-carboxamide groups of glutamine residues and free amino groups [17].

Glutaraldehyde is among the most widely used chemical cross-linking agents, and provides a high degree of cross-linking of protein–polysaccharide complexes [18]. Nevertheless, the use of high concentrations of the cross-linking agent is associated with the occurrence of adverse drug reactions, such as breathing disorders, development of asthma, and dermatitis, as well as irritation of the nasal mucosa and ocular conjunctiva [19]. 

Physical mixing of two biopolymers leads to the formation of non-covalent protein–polysaccharide bonds, which largely depends on the nature of the biopolymers, their quantitative ratio, pH of the medium, and ionic strength [20]. Regarding physical protein–polysaccharide complexes, non-covalent interactions can be due to hydrogen bonding and electrostatic and hydrophobic interactions [21]. Most polysaccharides, except for chitosan, are negatively charged, and at pH values below the isoelectric point of casein, the formation of a casein–polysaccharide complex occurs [22]. 

Several studies on the preparation of casein–polysaccharide complexes have been reported in the literature, but only a few are related to the preparation of composite casein–polysaccharide drug delivery nanocarriers. Markman et al. [23] reported the preparation of vitamin D2-loaded casein–maltodextrin nano capsules. The obtained conjugates show a high percentage of therapeutic agent incorporation efficiency (up to 90%). Furthermore, the carrier demonstrated high colloidal stability and prevention of vitamin D2 oxidation in acidic pH conditions. Regarding the release, the carrier showed a pH-dependent behavior. Lack of release of the incorporated nutraceutical was observed at an acidic pH (2.5), while at pH 7 up to 90% of the incorporated nutraceutical was released, demonstrating the potential use of the carrier as a drug delivery system for targeted delivery in the small intestine. In another study, a drug delivery system for epigal-locatechin-3-gallate was obtained via polyelectrolyte complexation between casein and chitosan [24]. The obtained nanostructures demonstrated an average particle size of 150 nm, a high degree of biocompatibility, and improved intestinal permeability of epi-gallocatechin-3-gallate. As a result of polyelectrolyte complexation, the preparation of alginate-coated casein nanoparticles loaded with doxorubicin has also been reported [25]. The results showed that, after coating the casein particles with alginate, an increase in their average particle size from 264 nm to 294 nm was observed. Furthermore, the drug delivery system showed a high drug loading index and encapsulation efficacy (95%) of the cytostatic agent—doxorubicin. In vitro release was performed in two different media (pH 7.4 and pH 5), simulating systemic circulation and tumor cell pH. The results demonstrated a delayed release in alkaline pH and a more pronounced release in acidic pH [25].

One of the polysaccharides with potential application for the preparation of casein–polysaccharide complexes is the biopolymer fucoidan. Fucoidan belongs to a family of sulfated, L-fucose-rich polysaccharides found in the cell wall of various species of brown algae (Phaeophyta: Laminariaceae, Fucaceae, Chordariaceae, and Alariaceae) [26]. It can also be obtained from sea cucumbers (Holothuroidea: Stichopodidae, Holothuri-idae), sea urchin eggs (Echinoidea: Strongylocentrotidae, Arbaciidae), and sea grasses (Cymodoceaceae) [27]. Over 90% of the total sugar content in fucoidan consists of deoxyhexose—L-fucose [28]. Along with fucose, other monosaccharides can be found in the fucoidan structure: galactose, glucose, xylose, mannose, rhamnose, arabinose, uronic acid, and acetyl groups [29]. Although the structure and composition of fucoidans differ depending on the origin and extraction method, fucoidans can be categorized into two main types according to their monomeric subunits (Figure 1).

Type I fucoidan is composed of an α-(1→3)-α-L-fucopyranose backbone linked to sulfate radicals at the C2 and C4 positions, while type II fucoidan is composed of an α-(1→3)- and α-(1→4)-α-L-fucopyranose backbone, with sulfate radicals attached at the C2, C3, and C4 positions [28,29].

Depending on the sulfate content and molecular weight, fucoidans show different biological effects (antioxidant, antibacterial, antiviral, anticoagulant, hypolipidemic, hypoglycemic, antitumor, anti-inflammatory, and immunomodulatory). In addition to its biological activity, fucoidan also possesses suitable physicochemical properties as a drug carrier, which is confirmed by numerous reports in the literature about micro- and nanoparticles developed on this basis, liposomes, films, hydrogels, etc. [30,31] The polymer backbone has anionic properties due to the presence of negatively charged sulfate ester groups. Depending on the specific chemical characteristics and charge density, fucoidan can interact with various biomolecules, such as proteins and other polysaccharides [32]. It is soluble in water and in acidic solutions, and its solubility depends on the molecular weight and the number of sulfate groups. The highly water-soluble fucoidans form solutions with low viscosity and are therefore not generally used as gelling agents. On the other hand, the viscoelastic properties of fucoidan are highly dependent on the origin of the polysaccharide, its concentration, molecular weight, sulfate content, pH, and temperature [27]. The interaction of fucoidan with oppositely charged biopolymers, such as chitosan [33], poly (2-hydroxyethyl methacrylate) [34], soy protein isolate (SPI) [35], and lactoglobulin [36], can lead to the formation of polyelectrolyte complexes with improved characteristics for biomedical application. Polyelectrolyte complexation (complex coacervation) is one of the most widely used methods for obtaining fucoidan micro- and nanoparticles. Other common techniques for formulating drug systems from the polysaccharide are simple coacervation, ionotropic gelation, spray drying, emulsification, etc. [37].

With the possibilities of nanotechnology, the development of innovative carriers capable of providing selective and specific therapy for several diseases has long been the subject of fundamental pharmaceutical science. Polymers of natural origin such as casein and fucoidan enable the development of biodegradable nanocarriers providing modified release and targeted delivery, which significantly improves the distribution of the incorporated drugs in the body and reduces the risk of adverse drug reactions. Their complex biological effects and the ability to form electrostatic interactions provide an opportunity for the development of casein–fucoidan composite nanoscale drug delivery systems.

The aim of the present work is to develop casein–fucoidan composite nanostructures by polyelectrolyte complexation and cross-linking of the complex with glutaraldehyde. By further applying 3(k-p) fractional factorial design, the work aims to evaluate the effect of dependent variables on independent variables and outline optimal technological and process parameters for the development of effective drug delivery systems.

## 2. Materials

Fucoidan from Macrocystis pyrifera (11.1 kDa), sodium caseinate (from bovine milk), and glutaraldehyde 25% (Mw 100.12 g/mol) were purchased from Sigma-Aldrich (St. Louis, MO, USA). All other reagents were of analytical grade.

## 3. Methods

### 3.1. Preparation of Casein/Fucoidan Composite Nanoparticles

Casein/fucoidan composite nanoparticles were obtained by polyelectrolyte complexation, cross-linking of the complex with glutaraldehyde, and subsequent spray drying (Figure 2).

Sodium caseinate was dissolved in deionized water pre-acidified with 0.1 M HCl solution to pH 2 to ensure protonation of the amino groups in the protein structure. For 15 min, the resulting solution was added dropwise under continuous stirring at 25,000 rpm (Miccra MiniBatch D-9, MICCRA GmbH, Heitersheim, Germany) to an aqueous fucoidan solution (pH 7), negatively charged due to the presence of sulfate esters groups in its structure. As a result of the interaction of the oppositely charged biopolymers, a polyelectrolyte complex was formed, which was further cross-linked with glutaraldehyde (0.004 µL of glutaraldehyde solution for 1 mL of casein solution). The obtained nanosuspension was spray dried under the following conditions: mesh size of the spray membrane 4.0 μm; inlet temperature 30 °C; solution feed rate 30%; sputtering intensity 40%; drying gas rate 120 L/min; pressure 30 nbar. To derive optimal production conditions and study the influence of the production parameters (casein/fucoidan ratio, glutaraldehyde concentration) and the technological parameter sputtering intensity on the size, yield, and zeta potential, a 3(k-p) partial experimental design was applied. The advantage of the 3(k-p) design over the full factorial design is that not all possible combinations between independent variables are considered, which greatly reduces the number of models to be developed and therefore shortens the run time [18]. Each of the independent variables was varied at three levels: low (−1), medium (0), and high (+1), resulting in nine possible combinations. The three levels of varying the ratio of casein to fucoidan were 1:1, 2:1, and 3:1, using 0.25% (*w*/*v*), 0.33% (*w*/*v*), and 0.37% (*w*/*v*) casein solutions and 0.13% (*w*/*v*), 0.17% (*w*/*v*), and 0.25% (*w*/*v*) fucoidan solutions. Glutaraldehyde concentration was 1% (*w*/*v*), 2% (*w*/*v*), and 3% (*w*/*v*) and the sputtering intensity was 30%, 40%, and 50%.

### 3.2. Characterization of Physicochemical Properties

#### 3.2.1. Production Yield

The production yields of the obtained nanoparticles from different batches were calculated according to Equation (1):(1)$$Production\;yield\left(\%\right)=\frac{Spray\;dried\;nanoparticles\left(mg\right)}{Casein\left(mg\right)+Fucoidan\left(mg\right)}\times 100$$

#### 3.2.2. Particle Size Analysis, Size Distribution, and Zeta Potential

The particle size of the obtained casein/fucoidan composite structures was analyzed by dynamic light scattering (Microtrac, York, PA, USA). The system measures the particle size in the range of 0.8 nm to 6.5 µm with non-invasive backscattering technology using a 3-mW helium/neon laser at a 780 nm wavelength. The system allows measurements of ζ-potential in the range from −200 mV to +200 mV. All analyses were performed in triplicates at 25 °C.

#### 3.2.3. Scanning Electron Microscopy

Visualization of the obtained structures was performed using scanning electron microscopy (Prisma E SEM, Thermo Scientific, Waltham, MA, USA). The samples were loaded on a copper sample holder and sputter coated with carbon followed by gold using a vacuum evaporator (Quorum Q150T Plus, Quorum Technologies, Lewes, UK). The images were recorded at 20 kV acceleration voltage, 10,000× magnifications using an ETD (Everhart–Thornley) detector.

#### 3.2.4. Transmission Electron Microscopy

The particle size and morphology of the structures were investigated by TEM (Talos F200X, Thermo Fisher Scientific, Waltham, MA, USA). Nanocomposite suspension was added dropwise onto a formvar/carbon-coated copper grid, and then the TEM observation of the samples was performed at an operating voltage of 200 kV.

#### 3.2.5. Fourier Transform Infrared Spectroscopy (FTIR)

The structural characteristics of casein, fucoidan, and the Cas/FN composite nanoparticles were characterized by FTIR. The spectra were collected in the range from 600 cm^−1^ to 4000 cm^−1^ with a resolution of 4 nm and 16 scans, using a Nicolet iS 10 FTIR spectrometer (Thermo Fisher Scientific, Pittsburg, PA, USA). The instrument is equipped with a diamond attenuated total reflection (ATR) accessory and the spectra were analyzed with OMNIC^®^ software package (Version 7.3, Thermo Electron Corporation, Madison, WI, USA).

#### 3.2.6. Differential scanning calorimetry (DSC)

The thermal stability and phase state of the substances in the composition of the final structures were investigated using a differential scanning calorimeter DSC 204F1 Phoenix (Netzsch Gerätebau GmbH, Selb, Germany). An indium standard (Tm = 156.6 °C, ΔHm = 28.5 J/g) was used for temperature and heat flux calibration. The sample for analysis was placed in an aluminum crucible and hermetically sealed, then heated in a temperature range of 25 °C to 300 °C at a heating rate of 10 °C/min. The results were processed by Proteus 8.0 software (Netzsch Gerätebau GmbH, Selb, Germany), and the obtained thermograms reflect the variation of the heat flux relative to the mass of the sample as a function (mW/mg) of the temperature.

### 3.3. Experimental Design and Statistical Analysis

To determine the optimal conditions for the preparation of the composite structures and to study the influence of the production and technological parameters over the main structural and morphological characteristics of the particles, a 3(k-p) fractional factorial design was applied.

Three optimal experimental responses were studied: production yield (Y1), average particle size (Y2), and zeta potential (Y3). They were the results of the individual influence and the linear interactions of the three independent variables. Hence, the responses were modeled by the following linear polynomial model:Y = β0 + β1X1 + β2X2 + β3X3 + β4X1 X2 + β5X1 X3 + β6X2 X3

The results of these experiments were compared using analysis of variance (ANOVA), which was able to determine if the factors were significant. To test whether the terms were significant in regression model, t-tests were performed using a 95% (α = 0.05) level of significance. Furthermore, F-test was used to determine if there was an overall regression relationship between the response variable Y and independent variables X1, X2, X3 and their linear interactions at a 95% (α = 0.05) level of significance. The statistical analysis was performed using Statistica 8 (TIBCO Software Inc., Palo Alto, CA, USA).

## 4. Results

### 4.1. Preliminary Experiments

Since the proposed method for the preparation of the structures is unique, it was necessary to define the technological parameters for the preparation of the composites by the method of spray drying.

#### 4.1.1. Optimal Spray Head Mesh Size

To determine the optimal spray head mesh size, a suspension containing casein–fucoidan composite nanostructures obtained at a casein:fucoidan ratio of 1:1 and cross-linked with 1% glutaraldehyde solution was sprayed through three spray mesh heads with a pore size of 4 µm, 5.5 µm, and 7 µm, at inlet temperature 30 °C, solution feed rate 30%, spray intensity 30%, and drying gas rate 120 L/min. The results showed that the increase in the spray mesh head leads to an increase in the particle size of the obtained structures. Since the aim was to obtain particles with sizes between 100 nm and 500 nm, 4 μm was chosen as the optimal spray head mesh size, providing the formation of composite particles with an average size of 137.22 ± 21 nm.

#### 4.1.2. Inlet Temperature and Solution Feed Rate

Since casein denatures at temperatures above 60 °C, to maintain the stability of the protein a temperature of 30 °C was chosen as optimal for the spray drying.

To determine the optimum solution feed rate of the process, a suspension containing casein–fucoidan composite nanostructures prepared at a casein:fucoidan ratio of 1:1, cross-linked with 1% glutaraldehyde, and spray dried at an inlet temperature of 30 °C, spray mesh head 4 µm, spray intensity of 30%, gas flow rate 120 L/min, and solution feed rate varying from 30% to 60%. The results showed that spraying at 20% solution feed rate led to the formation of smaller particles, 122 ± 27 nm, but the spray drying time was prolonged. Spray drying at a solution feed rate of 30% and 40% resulted in the production of structures with an average particle size of 147 ± 28 nm and 388 ± 21 nm. Further increase in the solution feed rate up to 50% resulted in the production of larger particles with an average size of 602 ± 35 nm. Based on the obtained results, the optimum solution feed rate to was set to 30%.

#### 4.1.3. Gas Flow Rate

The optimum gas flow rate was found in the atomization of a suspension containing casein–fucoidan composite nanostructures prepared at a casein:fucoidan ratio of 1:1, cross-linked with 1% glutaraldehyde, and spray dried at an inlet temperature of 30 °C, spray mesh head 4 µm, spray intensity of 30%, solution feed rate 30%, and varying gas flow rate from 90 L/min to 120 L/min. The results showed that spray drying at a gas flow rate of 90 L/min was inefficient and resulted in deposition of the polymer to the walls of the glass chamber. Increasing the gas flow rate to 100 L/min led to effective drying of a small part of the feeding suspension, but at the same time a part of the polymer was deposited on the walls of the glass chamber. Further increase of the gas flow rate to 120 L/min resulted in efficient drying of the suspension and the formation of composite structures demonstrating a significant yield (64.8 ± 3.0) %. As a result of these preliminary tests, a gas flow rate of 120 L/min was selected as optimal for the spray drying of the composites.

#### 4.1.4. Spray Intensity

Based on the literature review regarding the technological parameters affecting the process of nano spray drying, it was found that the spray intensity largely affects the main characteristics of the obtained particles. Therefore, the influence of this technological parameter on the structural-morphological characteristics of the obtained particles was determined by applying appropriate factorial design.

### 4.2. Experimental Design

To determine the optimal conditions and to study the influence of the production parameters (casein/fucoidan ratio, glutaraldehyde concentration) and the technological parameter spray intensity on the size, yield, and zeta potential, a 3(k-p) fractional factorial design was applied. The dependent variables were production yield (Y1), average particle size (Y2), and zeta potential (Y3), and the independent variables were casein:fucoidan ratio (X1), glutaraldehyde concentration (X2), and spray intensity (X3) (Table 1). Each of the independent variables was varied at three levels: low (-1), medium (0), and high (+1), resulting in nine possible combinations (Table 2).

To estimate the effects of the independents on the dependent variables and to generate polynomial equations for predicting the response, linear regression analysis and analysis of variance (one way ANOVA) were applied using Statistica 8 (TIBCO Software Inc., CA, USA). The main effects of the independent variables, which represent the mean values reflecting the change of an independent variable from its lowest to its highest level on the dependent variables, are presented in Table 3.

The main regression coefficients presented in the table, which apply to the linear equation, can be used to predict the responses at different levels of variation.
Y = β0 + β1X1 + β2X2 + β3X3 + β4X1 X2 + β5X1 X3 + β6X2 X3

β0 represents the intercept with the y-axis; X1, X2, X3 are the observed effects of the independent variables; X1 X2, X1 X3, and X2 X3 are the observed effects of the interaction, while β1 to β6 demonstrate the linear regression coefficients. After the evaluation of the main effects, the determination of the significant independent variables for each dependent variable is performed by analysis of variation (one way ANOVA) (Table 4).

In the table, the sum of squares (SS) provides information for estimating the main effects of the factors, and the F-values represent the ratios of the corresponding mean squared effects and mean squared errors [38]. The table also presents the *p*-values that show the statistical significance of each factor (*p* < 0.05).

Based on the data obtained from the analysis of variations, in terms of F- and p-values, the statistical significance of each of the independent variables, as well as their combined interaction on the investigated dependent variables, can be determined. The independent variables that demonstrate the highest F-values and *p* < 0.05 are the main factors with a statistically significant impact on the studied dependent variables [38].

The results show that the linear increase in casein/fucoidan ratio and spray intensity, and the linear interaction between casein/fucoidan and glutaraldehyde, as well as the interaction between glutaraldehyde and spray intensity, have a statistically significant effect on production yield (*p* < 0.05). Regarding particle size and zeta potential, only a linear increase in the casein/fucoidan ratio has a statistically significant effect (*p* < 0.05). The standardized effects of the independent variables are represented by Pareto charts, which rank the effects of the independent variables on the dependent variables in terms of their significance.

#### 4.2.1. Effect of Independent Variables on Particle Yield

The standardized effects of the independent variables are represented by the Pareto chart (Figure 3), which confirms the results of the analysis of variance (one way ANOVA) and shows that casein/fucoidan ratio (X1), spray intensity (X3), linear interaction between casein/fucoidan ratio, and glutaraldehyde concentration (X1X2), as well as the linear interaction between glutaraldehyde concentration and spray intensity (X2X3), have a statistically significant effect on the yield (*p* > 0.05) (Table 4).

Particle yield varied from (5.21 ± 2.2) % to (66.41 ± 2.2) % (Table 2). The model suggested that increasing the casein/fucoidan ratio and increasing the spray intensity led to a decrease in the yield. A similar antagonistic effect on yield was also observed with the simultaneous increase in the casein/fucoidan ratio and spray intensity. Furthermore, the model suggested that an increase in glutaraldehyde concentration and spray intensity led to an increase in the particle yield, as shown in Table 4. The two-way linear interaction between X1 and X2, as well as X2 and X3, demonstrated the change in the response that occurred when two independent variables were changed simultaneously. Positive coefficients indicated a synergistic effect and negative coefficients indicated an antagonistic effect on the response [39].

The influence of the statistically significant independent variables, as well as their complex influence on the yield, is further presented through 3D surface plots (Figure 4). The results prove that there is a tendency to increase the yield from (53.72 ± 1.9) % to (66.4 ± 2.2) %, when decreasing casein/fucoidan from 3:1 to 1:1 and decreasing glutaraldehyde concentration from 3% (*w*/*v*) to 1% (*w*/*v*) (Figure 4a). Probably, at higher concentrations of casein and glutaraldehyde, the viscosity of the feed suspension increases, which leads to an inefficient spray drying [40].

On the other hand, as the spray intensity decreases from 40% to 30% and the glutaraldehyde concentration increases from 1% (*w*/*v*) to 3% (*w*/*v*), there is a tendency toward an increase in the yield from (38.03 ± 1.8) % to (48.91 ± 2.9) % (Figure 4b). Probably, under these conditions, an effective cross-linking of the polyelectrolyte complex occurs, reducing the viscosity of the feed suspension, which in turn leads to its effective atomization at 30%. Increasing the spray intensity leads to the generation of a larger number of droplets per unit time, which coalesce and form irregularly shaped structures [40,41].

A linear regression analysis is applied to examine the fit of experimental to predicted data obtained in terms of particle yield [42]. A linear polynomial equation is generated based on the yield regression coefficients:Yield = 207.54 − 2.686.X1 − 6.4087.X2 − 0.4995.X3 + 0.3075X1.X2 + 0.0059X1.X3 − 0.3188.X2.X3

A correlation is observed between the experimental and the predicted data, which is confirmed by the coefficient of determination of the linear polynomial equation (R^2^ = 0.99). The value for the adjusted coefficient of determination (R^2^adj = 0.99) confirms that 99% of the obtained data can be explained by the model. The observed experimental data as well as the data predicted by the model are presented in Table 5.

One way ANOVA (one way ANOVA) was applied to estimate the significance of the model and its application for predicting the response at different levels of variation of the independent variables (Table 6). The obtained data present a significant model describing the influence of the independent variables, values Prob > F value and *p* < 0.05 (*p* = 0.002).

For the developed model, the possibility of observing constant variation and normal distribution of errors is assumed. To verify this claim, a normal probability plot of the residuals is generated in Figure 5a. The residuals are the differences in values between the experimental and predicted values of the response. The closer the residuals are to the theoretically expected, the more reliable the model is for predicting the response. From Figure 5a it can be clearly observed that all the experimental values are close to the theoretically expected (Figure 5a). Figure 5b is a plot representing the residuals versus the predicted response values. The results presented in Figure 5b show that the residuals are symmetrically located and tend to cluster in the low values of the Y-axis. Based on the obtained data, it can be assumed that the proposed model is adequate in terms of its ability to predict the response at different levels of variation of the independents.

#### 4.2.2. Effect of Independent Variables on Particle Size

The standardized effects of the independent variables on particle size are represented by the Pareto chart (Figure 6), which confirms the results from the analysis of variance (Table 4) and shows that only the casein/fucoidan ratio has a statistically significant effect on the yield of the nanoparticles (*p* > 0.05), while the remaining variables (X1X2) and their combined effects (X1X2, X1X3, X2X3) do not have a statistically significant effect on particle size.

The relationship between the independent and dependent variables is further represented by 3D surface plots (Figure 7). The figure visualizes the trend of the decrease in particle size from (1.515 ± 0.35) µm to (0.265 ± 0.03) µm with the decrease in the casein/fucoidan ratio from 3:1 to 1:1 and decrease in the glutaraldehyde concentration from 3% (*w*/*v*) to 1% (*w*/*v*). On one hand, probably, when equivalent amounts of the two biopolymers are used, an effective polyelectrolyte complexation occurs leading to the formation of more dense particles [43]. On the other hand, low concentrations of glutaraldehyde are sufficient to saturate the free functional groups and ensure effective covalent cross-linking of the complex and the formation of smaller composite nanostructures [44].

The results were confirmed by the performed scanning and transmission electron microscopy. On the SEM micrographs (Figure 8A) it can be noticed that the particles obtained at 1:1 casein/fucoidan ratio have smooth surface and a spherical shape, and their size ranged from (0.264 ± 0.03) µm (C1F1G1Sp.30) to (0.356 ± 0.01) µm (C1F1G2Sp.40). The observations were also confirmed by transmission electron microscopy (Figure 8B), where spherical particles of a size around 200 nm were found. Increasing the casein/fucoidan ratio resulted in the formation of particle agglomerates with irregular shape and size ranging from 0.842 ± 0.26 µm (C3F1G2Sp.30) up to 1.04 ±0.98 µm (C3F1G1Sp.40) and 1.51 ± 0.35 µm (C3F1G3Sp.50). Probably, increasing the concentration of casein leads to inefficient polyelectrolyte complexation between the polymers, and an inefficient amount of glutaraldehyde needed for the formation of denser particles. This leads to an increase in the viscosity of the feeding suspension and an inefficient spray drying process. 

Based on the obtained regression coefficients, a linear polynomial equation is derived as follows:Particle size = 0.34 − 0.0164X1 − 0.6813X2 − 0.0471X3 − 0.0003X1.X2 + 0.0004X1.X3 − 0.0187X2.X3

A good correlation is found between the experimental and predicted data (R^2^ = 0.81). A value of R^2^adj = 0.69 confirms that 69% of the obtained data can be explained by the model. The observed and predicted values are presented in Table 7.

Regarding the ability of the model to predict the response at different levels of variation of the independent variables, one way ANOVA is performed. Prob > F-value and *p* < 0.05 (*p* = 0.029) prove the ability of the model to predict the response (Table 8). 

To examine the variance and normal distribution of errors, a normal probability plot of residuals and a plot of residuals versus predicted values are generated Figure 9. The results show that all the experimental values are close to the theoretically expected ones (Figure 9a). Furthermore, the residuals are symmetrically located and tend to cluster in the low Y-axis values (Figure 9b). The obtained data confirms that the developed model can be used to predict the response at different levels of variation of the independent variables.

#### 4.2.3. Effect of Independent Variables on Particle Zeta Potential

The standardized effects of the independent variables on the zeta potential of the particles are represented by Pareto chart (Figure 10), which shows that only the casein/fucoidan ratio has a statistically significant effect on the zeta potential of the nanoparticles (*p* > 0.05), while the other variables (X2 and X3) and their combined effect (X1X2, X1X3, X2X3) do not have a statistically significant effect.

The relationship between the independent and the dependent variables is further represented by 3D surface plots (Figure 11). The figure shows a trend of an increase of the zeta potential from (−25.71 ± 1.0) mV to (−1.03 ± 1.9) mV with an increasing casein/fucoidan ratio from 1:1 to 3:1, which is probably due to the ineffective coating of the casein particles with fucoidan [37]. The hypothesis is also confirmed by the larger size of these particles, which can be explained by the insufficient amount of fucoidan required for the stabilization of the obtained structures [37].

Based on the obtained regression coefficients regarding the zeta potential, a linear polynomial equation is derived as follows:

Zeta potential = −83.33 + 1.24X1 −7.02X2 + 0.37X3 – 0.04X1.X2 – 0.01X1.X3 + 0.24X2.X3

A good correlation between experimental and predicted data is established (R^2^ = 0.96), while the value of R^2^adj = 0.94 confirms that 94% of the obtained data can be explained by the model (Table 9).

Regarding the ability of the model to predict the response at different levels of variation of the independent variables, one way ANOVA is performed. Prob > F-value and *p* < 0.05 (*p* = 0.0004) prove the ability of the model to predict the response (Table 10).

Constant variation and normal distribution of errors is represented by a normal probability plot and a plot of the residuals versus the predicted values (Figure 12). The results show that all the experimental data are close to the theoretically expected ones (Figure 12a). Furthermore, the residuals are symmetrically located and tend to cluster in the low Y-axis values (Figure 12b). Based on the presented results, variation and normal distribution of errors are proven, which confirms the adequacy of the model.

Based on the obtained results, models C1F1G1Sp.30, C1F1G2Sp.40, and C1F1G3Sp.50, which have an average particle size ranging from (0.265 ± 0.03) µm to (0.357 ± 0.02) µm, yields varying from (48.9 ± 2.9) % to (66.4 ± 2.2) %, and zeta potential from (−20.12 ± 0.9) mV to (−25.71 ± 1.0) mV, were selected as optimal for further use as drug delivery system.

### 4.3. Determining the Compatibility and Physical State of Casein/Fucoidan Composite Nanoparticles

To study the physical state and to understand the protein denaturation before and after functionalization with fucoidan, differential scanning calorimetry (DSC) was applied (Figure 13). After heating, casein exhibited a single endothermic denaturation transition peak (Tp) at 102.7 °C due to its dehydration and changes in the molecular integrity of the protein [45].

Unlike casein, fucoidan shows two peaks—an endothermic peak at 78.8 °C and an exothermic peak at 208.9 °C. The thermal decomposition of fucoidan is a four-step process. The first endothermic peak occurs due to the removal of physically absorbed water. The second exothermic peak is in the temperature range 200 °C–275 °C, which represents the second and the third decomposition steps occurring due to devolatilization processes. The last decomposition step usually occurs in the range 375 °C–1000 °C (outside the temperature range of the current study) and corresponds to oxidative degradation of the polymer [46]. 

The decomposition path of the synthesized casein–fucoidan composites followed a different pattern compared to the parent components. The degradation of the composites occurred as a two-stage process with two endothermic peaks observed in the DSC thermograms. The peak at 81.2 °C corresponds to moisture loss. The second peak at 213.3 °C is in the temperature range of the second and the third decomposition stages of fucoidan. However, transition from exothermic into endothermic process occurred, which could be related to an interaction between the components and formation of a conjugate of casein with fucoidan [47]. 

To determine the interaction between casein and fucoidan molecules and the changes in the chemical structure during the formation of the Cas/FN composite nanostructures, FTIR analysis was conducted (Figure 14). Casein shows peaks at 1646 cm^−1^ in the amide I region and 1530 cm^−1^ in the amide II region, which could be assigned to the stretching of the carbonyl group (C=O) and to the symmetric stretching of N-C=O bonds, respectively [48].

The FTIR spectrum of fucoidan showed a strong absorption band at 1224 cm^−1^, possibly related to the O=S=O vibration of sulfuric acid (Figure 14), as the band at 849 cm^−1^ was the characteristic band of the sulfate bond of fucoidan. Furthermore, the band at 1058 cm^−1^ was attributed to the C-O-C stretching vibration of the glycosidic bond [49]. For the Cas/FN composite nanoparticles, a clear shift in the amide I region from 1646 to 1642 cm^−1^ and a shift in the amide II region from1530 to 1542 cm^−1^ were observed, due to the electrostatic interactions and hydrogen bonding between protein and polysaccharide [18] (Figure 15).

## 5. Conclusions

Within the goal of this study, a novel, one-of-its-kind technological process for the preparation of casein–fucoidan composite nanostructures combining polyelectrolyte complexation and spray drying was developed. This research demonstrated that the main physicochemical properties of the obtained structures can be manipulated by various process parameters. The influence of the independent productional and technological factors was studied and optimized with the application of suitable surface response experimental design (3(k-p)). The results from the mathematical analysis showed that the casein to fucoidan ratio and spray intensity seemed to be crucial for the production yield of the obtained structure. Regarding the particle size and zeta potential, they were mainly affected by the casein to fucoidan ratio used for the preparation of the complexes. Therefore, the statistical experimental design methodology has clearly shown its usefulness for the optimization of the productional and technological parameters for the preparation of this innovative complex. Based on the results from the experimental design, the models obtained at casein/fucoidan ratio 1:1, cross-linked with 1% (*w*/*v*), 2% (*w*/*v*), and 3% (*w*/*v*) glutaraldehyde solution, and spray dried at 30%, 40%, and 50% spray intensity (C1F1G1Sp.30, C1F1G2Sp.40 and C1F1G3Sp.50), demonstrating production yields varying from (48.91 ± 2.9) % to 66.41 ± 2.2, particle size ranging from (0.264 ± 0.03) µm to (0.356 ± 0.01) µm, and zeta potential from (−20.12 ± 0.9) mV to (−25.71 ± 1.0 mV) were selected as optimal models for further use as drug delivery systems for various bioactive substances.

## 6. Patents

Results from this work contributed to the development of Utility model: BG4536U1; Nikolay Zahariev, Bissera Pilicheva; Medical University of Plovdiv; COMPOSITE NANOPARTICLES OF CASEIN/FUCOIDAN AS A SYSTEM FOR DELIVERY OF DAUNORUBICIN HYDROCHLORIDE; Bulgaria; 11 August 2023.

## Figures and Tables

**Figure 1 polymers-16-01818-f001:**
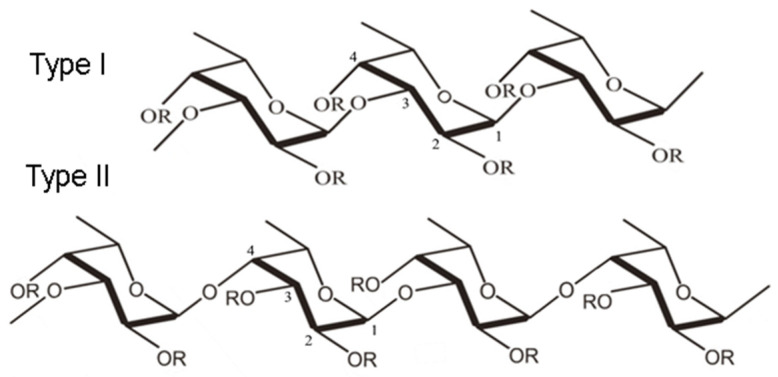
Fucoidan structure: α-(1→3)-α-L-fucopyranose skeleton linked with sulfate radicals at C2 and C4 positions (Type I) and α-(1→3)- and α-(1→4))-α-L-fucopyranose skeleton with sulfate radicals attached to C2, C3, and C4 positions (Type II) [27].

**Figure 2 polymers-16-01818-f002:**
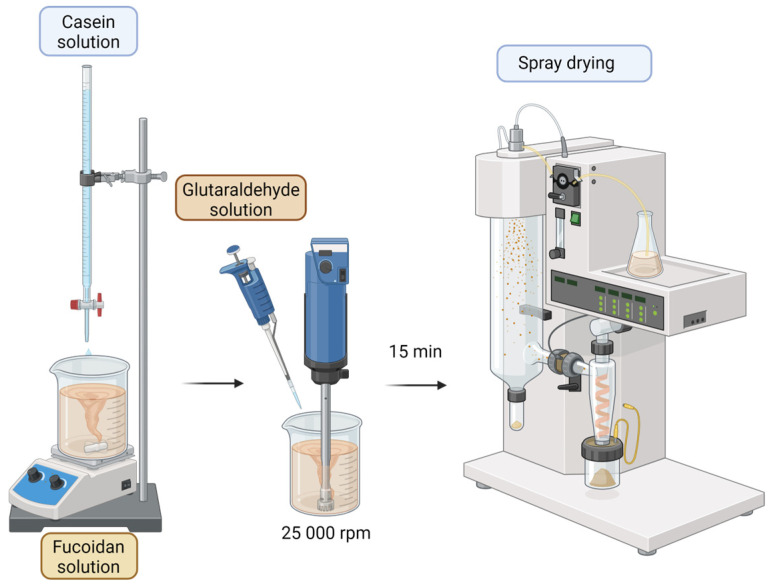
Schematic representation of the preparation of casein/fucoidan composite nanoparticles (Generated using BioRender™).

**Figure 3 polymers-16-01818-f003:**
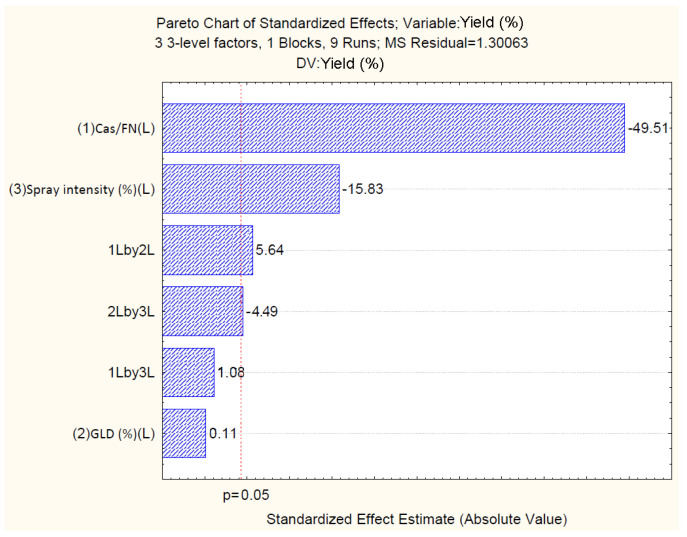
Pareto chart reflecting the standardized effect of independents on yield.

**Figure 4 polymers-16-01818-f004:**
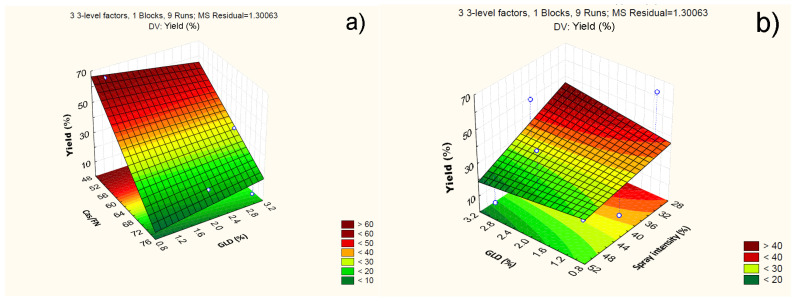
3D plot showing the effect of the casein to fucoidan ratio and glutaraldehyde concentration (**a**) and glutaraldehyde concentration and spray intensity (**b**) on particle yield.

**Figure 5 polymers-16-01818-f005:**
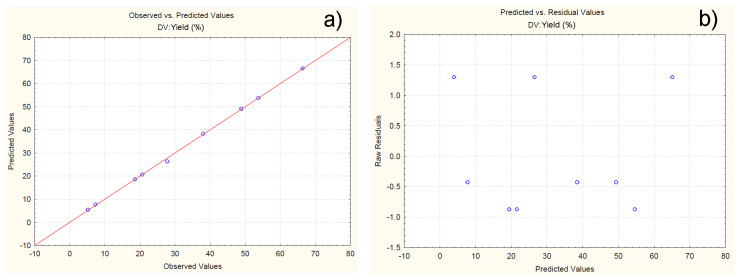
Normal probability plot of residuals with respect to yield (**a**); plot of residuals versus predicted values (**b**).

**Figure 6 polymers-16-01818-f006:**
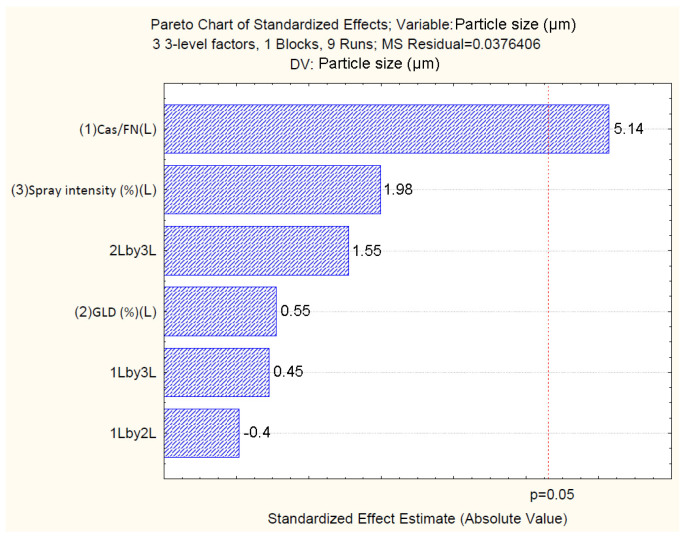
Pareto chart showing the standardized effect of independents on particle size.

**Figure 7 polymers-16-01818-f007:**
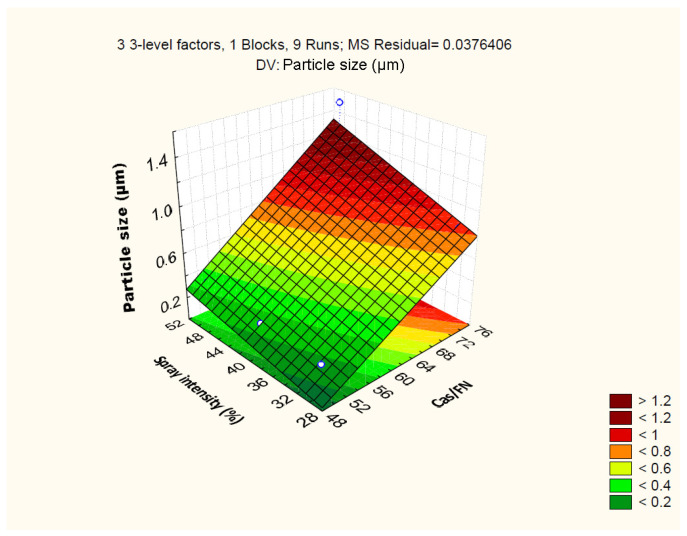
3D surface plot showing the effect of casein to fucoidan ratio and glutaraldehyde concentration on particle size.

**Figure 8 polymers-16-01818-f008:**
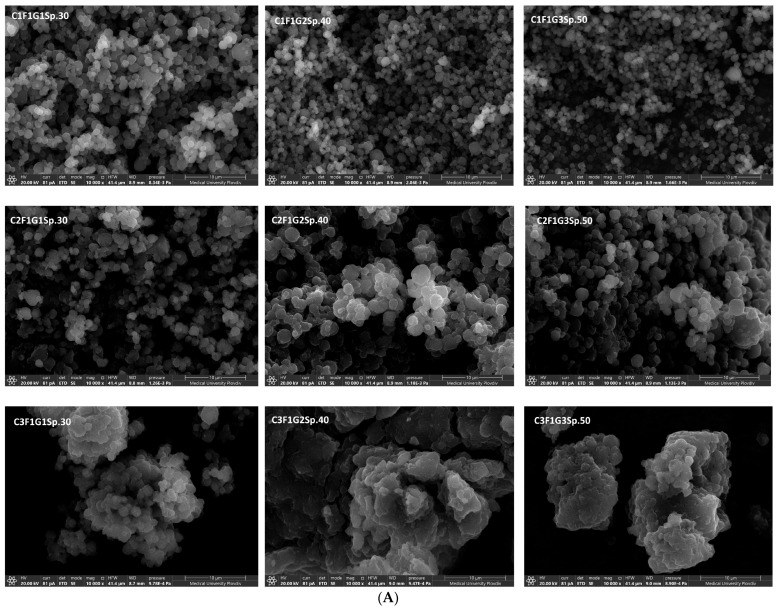
(**A**) SEM micrographs of casein/fucoidan composite nanostructures of different models obtained at 10,000× magnification, accelerating voltage 20 kV using an Everhart–Thornley detector. (**B**) TEM micrographs of casein/fucoidan composite nanostructures from model C1F1G1Sp.30 (squared section represents magnified area of the image).

**Figure 9 polymers-16-01818-f009:**
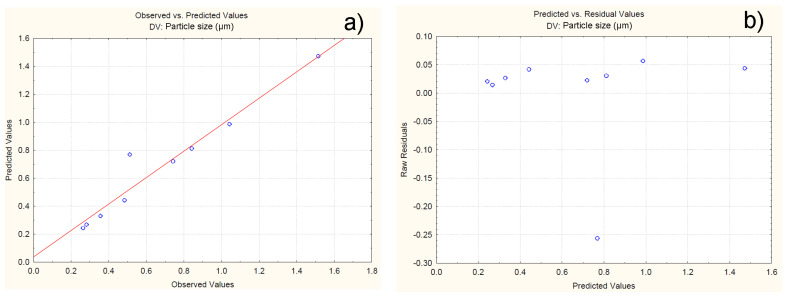
Normal probability plot of residuals with respect to size (**a**); plot of residuals versus predicted size values (**b**).

**Figure 10 polymers-16-01818-f010:**
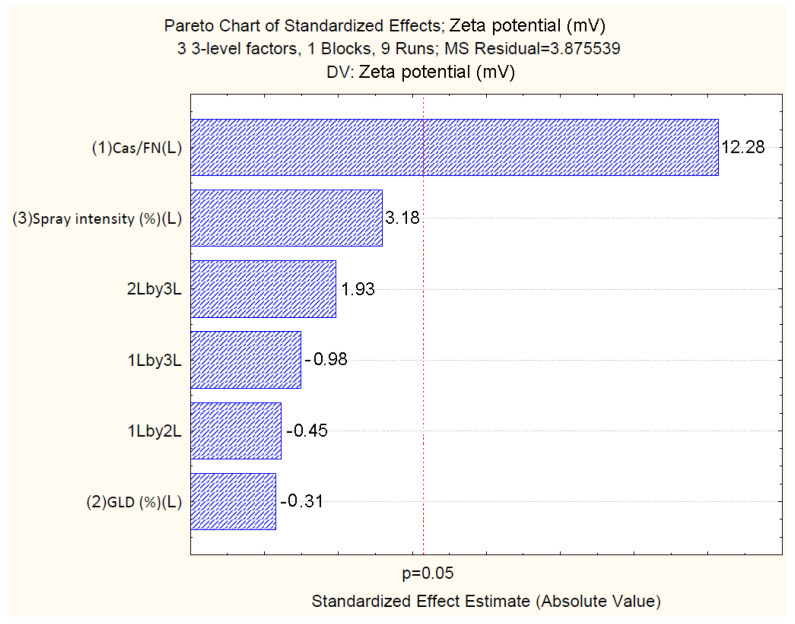
Pareto chart showing the standardized effect of independents on size.

**Figure 11 polymers-16-01818-f011:**
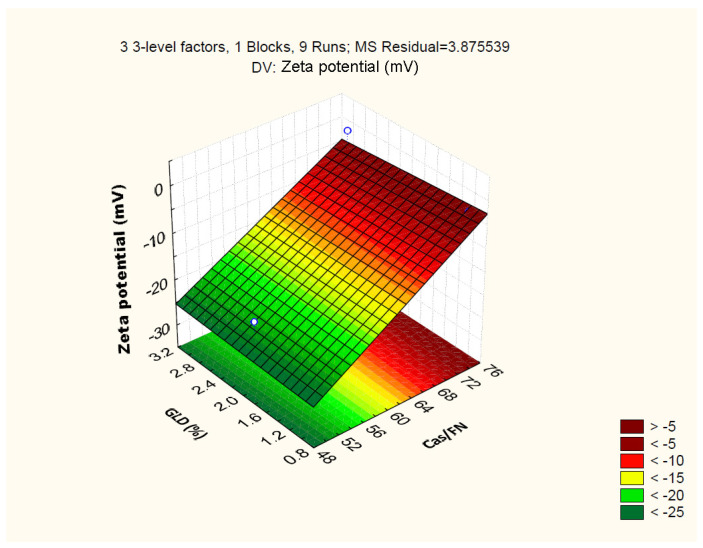
3D surface plot showing the influence of casein to fucoidan ratio and glutaraldehyde concentration on the zeta potential of the particles.

**Figure 12 polymers-16-01818-f012:**
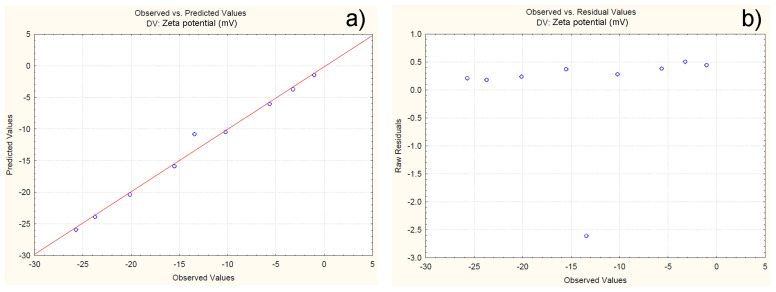
Normal probability plot of residuals versus zeta potential (**a**); plot of residuals versus predicted zeta values (**b**).

**Figure 13 polymers-16-01818-f013:**
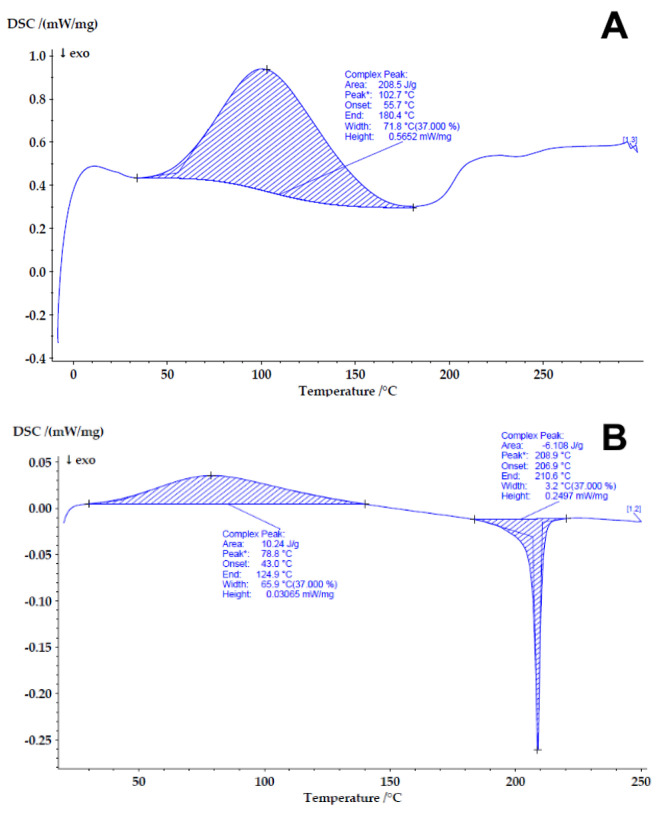
DSC thermograms of casein (**A**), fucoidan (**B**), and casein–fucoidan composite nanostructures (**C**).

**Figure 14 polymers-16-01818-f014:**
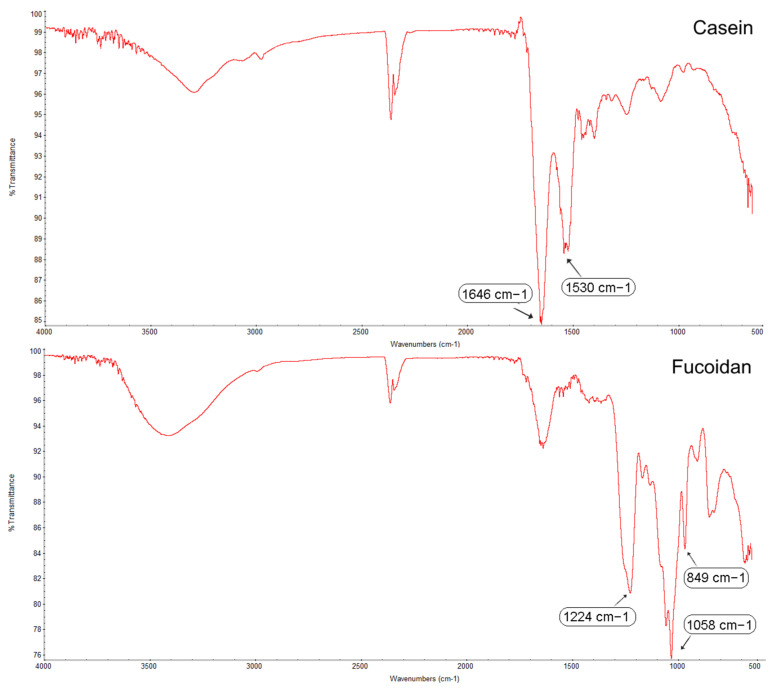
FTIR spectra of casein, fucoidan, and casein–fucoidan composite nanostructures.

**Figure 15 polymers-16-01818-f015:**
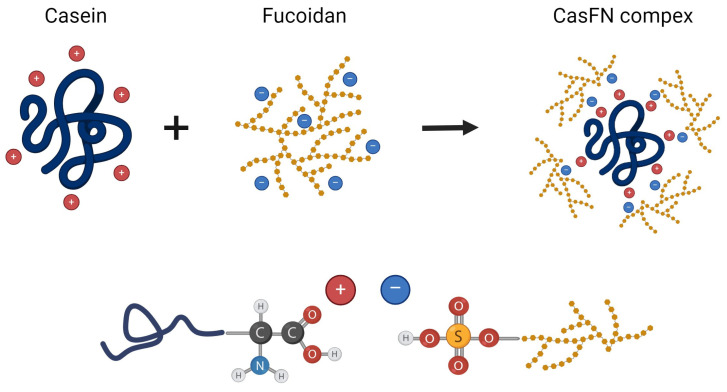
Schematic representation of the mechanism of formation of casein/fucoidan nanocomposites.

**Table 1 polymers-16-01818-t001:** Variables in the 3(k-p) experimental design.

Variables	Levels		
Independent Variables	−1	0	+1
X1: Casein:fucoidan ratio	1:1	2:1	3:1
X2: Glutaraldehyde concentration (%)	3	2	1
X3: Spray intensity (%)	30	40	50
Dependent Variables			
Y1: Yield (%)			
Y2: Particle size (µm)Y3: Zeta potential (mV)			

**Table 2 polymers-16-01818-t002:** Compositions and main characteristics of the nine nanoparticle models. **Cas/FN—casein to fucoidan ratio; GLD—glutaraldehyde concentration, Sp.i**nt.—spray intensity.

Models	Independent Variables			Dependent Variables		
	Cas/FN	GLD (%)	Sp.int. (%)	Size ± SD (µm)	ζ ± SD (mV)	Yield ± SD (%)
C1F1G1Sp.30	−1	−1	−1	0.264 ± 0.03	−25.71 ± 1.0	66.41 ± 2.2
C1F1G2Sp.40	−1	0	+1	0.356 ± 0.01	−20.12 ± 0.9	48.91 ± 2.9
C1F1G3Sp.50	−1	+1	0	0.282 ± 0.02	−23.72 ± 1.8	53.72 ± 1.9
C2F1G1Sp.50	0	−1	+1	0.742 ± 0.25	−10.21 ± 1.5	20.66 ± 1.5
C2F1G2Sp.40	0	0	0	0.512 ± 0.36	−13.43 ± 1.6	27.81 ± 2.6
C2F1G3Sp.30	0	+1	−1	0.484 ± 0.12	−15.51 ± 1.4	38.03 ± 1.8
C3F1G1Sp.40	+1	−1	0	1.043 ± 0.98	−3.23 ± 0.7	7.31 ± 3.4
C3F1G2Sp.30	+1	0	−1	0.842 ± 0.26	−5.65 ± 0.9	18.54 ± 2.7
C3F1G3Sp.50	+1	+1	+1	1.515 ± 0.35	−1.03 ± 1.9	5.21 ± 2.2

**Table 3 polymers-16-01818-t003:** Estimation of main effects of factors and coefficients for predictive mathematical models.

Factor	Yield (%)				Particle Size (µm)				ζ (mV)			
	Effect	SE	*p*	β	Effect	SE	*p*	β	Effect	SE	*p*	β
Mean/Int.	34.59	0.38	0.00	207.54	0.62	0.06	0.01	0.34	−14.35	0.66	0.00	−83.330
Cas/Fn (1)	−45.86	0.92	0.00	−2.68	0.80	0.15	0.03	0.01	19.64	1.59	0.00	1.244
GLD (2)	0.11	0.96	0.91	−6.40	0.09	0.16	0.63	−0.68	−0.15	1.67	0.78	−7.010
Sp.Int. (3)	−15.32	0.96	0.00	−0.49	0.32	0.16	0.18	−0.04	5.31	1.67	0.08	0.374
1 by 2	7.86	1.36	0.02	0.30	0.00	0.23	0.97	0.00	−1.08	2.35	0.69	−0.040
1 by 3	1.48	1.36	0.38	0.00	0.10	0.23	0.69	0.00	−2.32	2.35	0.42	−0.010
2 by 3	−6.37	1.41	0.04	−0.31	0.37	0.24	0.26	0.01	4.72	2.44	0.19	0.236

SE—standard error; *p*-value; β—regression coefficient.

**Table 4 polymers-16-01818-t004:** Summary results of the performed analysis of variance (one way ANOVA) regarding the influence of the independent variables and their combined influence on the dependent.

Factor	SS	df	MS	F	*p*
**Yield: R^2^ = 0.99; R^2^_Adj._ = 0.99; MS = 1.30**					
Cas/Fn (1)	3188.93	1	3188.93	2451.84	0.00
GLD (2)	0.01	1	0.01	0.01	0.91
Spray.Int (3)	326.21	1	326.21	250.81	0.00
1 by 2	41.43	1	41.43	31.86	0.02
1 by 3	1.54	1	1.54	1.18	0.38
2 by 3	26.25	1	26.25	20.18	0.04
Error	2.60	2			
Total SS	3632.52	8			
**Particle size: R^2^ = 0.94; R^2^_Adj._ = 0.77; MS = 0.03**					
Cas/Fn (1)	0.99	1	0.99	26.40	0.03
GLD (2)	0.01	1	0.01	0.30	0.63
Spray.Int (3)	0.14	1	0.14	3.95	0.18
1 by 2	0.00	1	0.00	0.00	0.97
1 by 3	0.00	1	0.00	0.20	0.69
2 by 3	0.09	1	0.09	2.40	0.26
Error	0.07	2			
Total SS	1.36	8			
**Zeta potential: R^2^ = 0.98; R^2^_Adj._ = 0.95; MS = 3.87**					
Cas/Fn (1)	585.15	1	585.15	150.98	0.00
GLD (2)	0.37	1	0.37	0.09	0.78
Spray.Int (3)	39.31	1	39.31	10.14	0.08
1 by 2	0.82	1	0.81	0.21	0.69
1 by 3	3.79	1	3.79	0.97	0.42
2 by 3	14.44	1	14.44	3.72	0.19
Error	7.75	2	3.87		
Total SS	633.89	8			

SS—sum of squares; df—degree of freedom; MS—mean sum of squares; F-value; *p*-value.

**Table 5 polymers-16-01818-t005:** Experimental and model-predicted values of relative yields of particles (%).

Batch	Observed	Predicted	Residuals	Error (%)
C1F1G1Sp.30	66.41	66.52	−0.11	−0.16
C1F1G2Sp.40	48.91	49.12	−0.21	−0.42
C1F1G3Sp.50	53.72	53.74	−0.02	−0.03
C2F1G1Sp.50	20.66	20.69	−0.03	−0.14
C2F1G2Sp.40	27.81	26.34	1.46	5.28
C2F1G3Sp.30	38.03	38.36	−0.33	−0.86
C3F1G1Sp.40	7.31	7.74	−0.45	−5.88
C3F1G2Sp.30	18.54	18.58	−0.04	−0.21
C3F1G3Sp.50	5.21	5.49	−0.24	−5.37

**Table 6 polymers-16-01818-t006:** ANOVA significance testing of the proposed model.

R^2^	R^2^_adj._	SS	df	MS	SS	df	MS	F	*p*
0.999	0.997	3629.92	6	604.98	2.601	2	1.300	465.148	0.002

**Table 7 polymers-16-01818-t007:** Experimental and model-predicted values of relative particle size (µm).

Batch	Observed	Predicted	Residuals	Error (%)
C1F1G1Sp.30	0.264	0.243	0.02	−0.21
C1F1G2Sp.40	0.356	0.329	0.02	−0.23
C1F1G3Sp.50	0.282	0.267	0.01	−0.17
C2F1G1Sp.50	0.742	0.719	0.02	−0.28
C2F1G2Sp.40	0.512	0.768	-0.25	2.60
C2F1G3Sp.30	0.484	0.442	0.04	−0.37
C3F1G1Sp.40	1.043	0.986	0.05	−0.50
C3F1G2Sp.30	0.842	0.811	0.03	−0.38
C3F1G3Sp.50	1.515	1.471	0.04	−0.44

**Table 8 polymers-16-01818-t008:** ANOVA significance testing of the proposed model.

R^2^	R^2^_adj_	SS	df	MS	SS	df	MS	F	*p*
0.810	0.696	1.102	3	0.367	0.258	5	0.0517	7.105	0.0297

**Table 9 polymers-16-01818-t009:** Experimental and model-predicted values of particle zeta potential (mV).

Batch	Observed	Predicted	Residuals	Error (%)
C1F1G1Sp.30	−25.71	−25.91	0.20	−0.81
C1F1G2Sp.40	−20.12	−20.35	0.23	−1.18
C1F1G3Sp.50	−23.72	−23.89	0.17	−0.75
C2F1G1Sp.50	−10.21	−10.49	0.28	−2.75
C2F1G2Sp.40	−13.43	−10.82	2.60	−19.4
C2F1G3Sp.30	−15.51	−15.88	0.37	−2.39
C3F1G1Sp.40	−3.23	−3.73	0.50	−15.63
C3F1G2Sp.30	−5.65	−6.03	0.38	−6.75
C3F1G3Sp.50	−1.03	−1.47	0.44	−43.05

**Table 10 polymers-16-01818-t010:** ANOVA significance testing of the proposed model.

R^2^	R^2^_adj._	SS	df	MS	SS	df	MS	F	*p*
0.9642	0.9642	611.22	3	203.74	22.66	5	4.533	44.94	0.00048

## Data Availability

The original contributions presented in the study are included in the article, further inquiries can be directed to the corresponding authors.

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
