# Peer review of "A Novel Method for the Preparation of Casein–Fucoidan Composite Nanostructures"

_polymers, 2024, doi:10.3390/polym16131818_

Round 1
Reviewer 1 Report
Comments and Suggestions for Authors
Author Response
The comments for the manuscript entitled, A novel method for the preparation of casein fucoidan composite nanostructures are given in the attached file.

Reviewer 2 Report
Comments and Suggestions for Authors
1. Authors themselves are stressing on the “ability to form smaller in size particles [3]” but in this study they are mentioning in the abstract to synthesise nanoparticles that “have an average particle size ranging from (265 ± 32) nm to 17 (357 ± 25) nm”. Is it not a very big size to be said to NPs?
2. Table 2 indicates very large particle size (in micro-meter) and yield is also less than 50% in most of the cases. Is it worth to consider such a method of preparation?
3. In Fig. 3, why the yield was fixed to be >0.05
4. Data given in table 5 shows “Experimental and model-predicted values of relative yields of particles (%)” and there is an unexpected matching between two. How it was so accurate theoretical prediction and experimental formation?
5. Fig. 5 b is missing something.
6. Similarly check Fig 9 b and 12 c
7. Paper is well written and contains good information.
8. However, justification of present materials to prepare NPs as well as the suitability present methodology should be compared with other similar studies available in the literature. Moreover, a comparison between the other materials and methods should be given.
9. Thus, paper needs a major revision.
Author Response
The comments for the manuscript entitled, A novel method for the preparation of casein fucoidan composite nanostructures are given below

Round 2
Reviewer 1 Report
Comments and Suggestions for Authors
Many of my previous comments are not rectified.
It is recommended to include TEM in the main manuscript. And provide details of which sample TEM was analyzed?
DSC of fucoidan (exothermic peak) and CAS/FN composite (two endotherms) are not explained properly.
To determine the thermal stability of the compounds, TGA should be analyzed.
Plausible chemical structure for the CAS/FN composite should be included for proper interpretation of FTIR.
Author Response
The authors appreciate the reviewer's valuable comments and remarks. The responses are presented below:
Comment 1: It is recommended to include TEM in the main manuscript. And provide details of which sample TEM was analyzed?
Response: TEM image was included in the manuscript.
Comment 2: DSC of fucoidan (exothermic peak) and CAS/FN composite (two endotherms) are not explained properly.
Response: We have revised the discussion of the results (marked in yellow).
Comment 3: To determine the thermal stability of the compounds, TGA should be analyzed.
Response: We accept the reviewer's remark. However, thermal stability was not a goal of this study. We will consider TGA analysis in our future work on this project.
Comment 4: Plausible chemical structure for the CAS/FN composite should be included for proper interpretation of FTIR.
Response: We have added a schematic representation of the mechanism of interaction of the components.
Reviewer 2 Report
Comments and Suggestions for Authors
The paper in its revised form is recommended for acceptance.
Author Response
The authors appreciate the reviewer's valuable time to evaluate the manuscript.
We are grateful for the reviewer's positive opinion.
Round 3
Reviewer 1 Report
Comments and Suggestions for Authors
The authors have kept SEM image instead of TEM image in Figure 8b. Replace it with TEM image.
Author Response
We have taken into account the reviewer's remark and we have added a TEM picture obtained by Talos F200 transmission electron microscope in the revised version of the manuscript (figure 8b).